# Exposure to *Coxiella burnetii* in Wild Lagomorphs in Spanish Mediterranean Ecosystems

**DOI:** 10.3390/ani14050749

**Published:** 2024-02-28

**Authors:** Sabrina Castro-Scholten, Javier Caballero-Gómez, David Cano-Terriza, Débora Jiménez-Martín, Carlos Rouco, Adrián Beato-Benítez, Leonor Camacho-Sillero, Ignacio García-Bocanegra

**Affiliations:** 1Grupo de Investigación en Sanidad Animal y Zoonosis (GISAZ), Departamento de Sanidad Animal, UIC Zoonosis y Enfermedades Emergentes ENZOEM, Universidad de Córdoba, 14014 Córdoba, Spain; sabrina1996cs@gmail.com (S.C.-S.); davidcanovet@gmail.com (D.C.-T.); debora.djm@gmail.com (D.J.-M.); adrianbeato.vet@gmail.com (A.B.-B.); v62garbo@uco.es (I.G.-B.); 2Grupo de Virología Clínica y Zoonosis, Unidad de Enfermedades Infecciosas, Instituto Maimónides de Investigación Biomédica de Córdoba (IMIBIC), Hospital Universitario Reina Sofía, Universidad de Córdoba, 14004 Córdoba, Spain; 3CIBERINFEC, ISCIII—CIBER de Enfermedades Infecciosas, Instituto de Salud Carlos III, 28029 Madrid, Spain; 4Departamento Biología Vegetal y Ecología, Área de Ecología, Universidad de Sevilla, 41012 Sevilla, Spain; c.rouco@gmail.com; 5Programa de Vigilancia Epidemiológica de la Fauna Silvestre en Andalucía (PVE), Consejería de Sostenibilidad, Medio Ambiente y Economía Azul, Junta de Andalucía, 29002 Málaga, Spain; leonorn.camacho@juntadeandalucia.es

**Keywords:** Q fever, wild rabbit, Iberian hare, One Health, risk factors

## Abstract

**Simple Summary:**

*Coxiella burnetii*, the causative agent of Q fever, is a multi-host zoonotic bacterium of public and animal health concern, with Spain being the European country with the highest number of Q fever cases in recent years. However, and despite that the European wild rabbit and the Iberian hare are two keystone species in the Iberian Peninsula and are considered important sources of food for humans, very little is known about the epidemiological role of these two species. To tackle this question, a cross-sectional study was carried out to determine the seroprevalence and risk factors associated with *C. burnetii* exposure in wild lagomorph populations of Southern Spain. Antibodies against this bacterium were found in 11.3% of 471 wild rabbits, and, for the first time, in 2.4% of 167 Iberian hares, which provides evidence of the moderate circulation of *C. burnetii* in wild lagomorph populations in Spanish Mediterranean ecosystems. Our results also demonstrated that wild lagomorphs from western Andalusia and those from hunting grounds in which sheep were present were at higher risk of exposure to *C. burnetii*. In this context, these risk factors should be prioritized in future risk-based surveillance programs for this zoonotic multi-host bacterium.

**Abstract:**

*Coxiella burnetii* is an important zoonotic pathogen of worldwide distribution that can infect a wide range of wild and domestic species. The European wild rabbit (*Oryctolagus cuniculus*) can play a role as a reservoir for this bacterium in certain epidemiological scenarios, but, to date, a very limited numbers of large-scale serosurveys have been conducted for this species worldwide. Although exposure in hare species has also been described, *C. burnetii* in Iberian hare (*Lepus granatensis*) has never been assessed. Here, we aimed to determine the seroprevalence and risk factors associated with *C. burnetii* exposure in wild lagomorphs in the Mediterranean ecosystems of southern Spain. Between the 2018/2019 and 2021/2022 hunting seasons, blood samples from 638 wild lagomorphs, including 471 wild rabbits and 167 Iberian hares, were collected from 112 hunting grounds distributed across all eight provinces of Andalusia (southern Spain). The overall apparent individual seroprevalence was 8.9% (57/638; 95% CI: 6.8–11.4). Antibodies against *C. burnetii* were found in 11.3% (53/471; 95% CI: 8.4–14.1) of the wild rabbits and 2.4% (4/167; 95% CI: 0.1–4.7) of the Iberian hares. Seropositive animals were detected for 16 (14.3%; 95% CI: 7.8–20.8) of the 112 hunting grounds tested and in all the hunting seasons sampled. A generalized estimating equations model showed that the geographical area (western Andalusia) and presence of sheep were risk factors potentially associated with *C. burnetii* exposure in wild lagomorphs. A statistically significant spatial cluster (*p* < 0.001) was identified in the south-west of Andalusia. Our results provide evidence of moderate, endemic and heterogeneous circulation of *C. burnetii* in wild lagomorph populations in Spanish Mediterranean ecosystems. Risk-based strategies for integrative surveillance programs should be implemented in these species to reduce the risk of transmission of the bacterium to sympatric species, including humans.

## 1. Introduction

The agent that causes Q fever disease, *Coxiella burnetii* (family *Coxiellaceae*), is an important and highly environmentally resistant zoonotic bacterium with a worldwide distribution [1]. In recent years, Europe has reported over 700 annual human cases of Q fever. Among European countries, Spain has recorded the highest number of Q fever cases in recent years [1], making it the most frequent reportable zoonosis in the country [2]. Different modes of transmission have been reported and although inhalation is considered the main mode, oral and tick-borne transmissions have also been evidenced [3]. In humans, Q fever causes fever, malaise, headache, muscle pain and endocarditis [4], whereas in animals, and particularly ruminants, this disease is characterized by abortions during late pregnancy or weak offspring, causing significant economic losses in the livestock industry [5]. 

Although domestic ruminants are the main reservoirs of the bacterium, a broad range of wild species can be infected with *C. burnetii*. The European wild rabbit (*Oryctolagus cuniculus*) has been denoted as a natural reservoir of *C. burnetii* in certain epidemiological contexts [6], and Q fever cases in humans have already been associated with indirect contact with wild rabbits [7]. Moreover, some studies have pointed out that hares may play a role in the maintenance and transmission of this zoonotic bacterium [8,9]. Nevertheless, information on the role of these lagomorph species in the epidemiology of Q fever is still very limited. 

The European wild rabbit and the Iberian hare (*Lepus granatensis*) are two endemic and keystone species of the Iberian Peninsula [10,11], being a source of food for a large number of predators [12]. In Mediterranean ecosystems, both species have been shown to be natural reservoirs for a wide range of pathogens that can affect other species, including humans [13,14,15,16]. This, together with their wide distribution, gregarious behavior and close direct and indirect contact with other sympatric species, evidenced the potential role of wild lagomorphs in the maintenance and transmission of multi-host pathogens such is *C. burnetii* [17]. However, even though the European Food Safety Authority (EFSA) has included Q fever as a priority for the establishment of a coordinated surveillance system [18] and has highlighted the need for the epidemiological surveillance of *C. burnetii* in wild lagomorphs to assess the circulation of this zoonotic pathogen [17], no or only a very few serosurveys have been conducted to date for Iberian hares and European wild rabbits, respectively, worldwide. Here, we aimed to assess the seroprevalence and risk factors associated with *C. burnetii* exposure in European wild rabbit and Iberian hare populations in Mediterranean ecosystems of southern Spain. 

## 2. Materials and Methods

### 2.1. Study Design and Sampling

A cross-sectional study was carried out on wild lagomorph populations in the region of Andalusia (south-western Europe) (87,268 km^2^; 36° N–38°60′ N, 1°75′ W–7°25′ W) between the 2018/2019 and 2021/2022 hunting seasons. The study area is characterized by a continental Mediterranean climate with mild wet winters and hot, dry summers. The western region presents higher mean humidity and less-extreme mean temperatures than the central and eastern regions [19].

The sample size was determined based on an assumed prevalence of 50%, which provides the maximum sample size in studies where the prevalence is unknown. The calculation was completed with a 95% confidence interval (95% CI) and a desired precision of ±5%. Whenever possible, 60 European wild rabbits were sampled in each of the eight provinces comprising the study area, in order to ensure a 95% probability of detecting at least one positive animal, assuming a minimum prevalence of 5% [20]. Sampling sites (hunting grounds) were randomly selected in each province. On each of these hunting grounds, hunters provided between 5 and 25 (mean: 12.1) European wild rabbits for sampling. A total of 471 wild rabbits from 38 hunting grounds distributed across all eight provinces were sampled during the study period. In addition, 167 Iberian hares from 82 hunting grounds were also sampled in the same study area and study period using a convenience sampling. In eight of these hunting grounds, both wild rabbit and Iberian hares were sampled.

Blood samples from all animals were obtained from the heart or thoracic cavity and centrifuged at 400× *g* for 10 min. The serum obtained was stored at −20 °C until serological analysis were performed. During sampling, an epidemiological questionnaire was also conducted through a direct interview with gamekeepers at each hunting grounds, wherever possible. The information obtained included the characteristics of the hunting ground, the presence of disease and control measures, management practices, and the presence of other sympatric species. Also, meteorological information for each sampling area [mean and maximum annual temperatures (°C), humidity (g/m^3^), and mean annual rainfall (mm)] was collected from the closest official meteorological station [19]. In addition, individual information, including species, location, year of sampling, age, kidney fat index and sex, was recorded for each animal. Bodyweight and body length were used as indicators of age [21]. 

### 2.2. Laboratory Analysis 

Serum samples were tested for antibodies against *C. burnetii* using the commercial indirect and multispecies enzyme-linked immunosorbent assay (ELISA) ID Screen^®^ Q Fever (IDvet, Grabels, France), according to the manufacturer’s instructions. This assay has previously been used in wild lagomorphs, being a satisfactory alternative to detect *C. burnetii* specific antibodies, compared to other commercial ELISA kits [22]. Results were expressed as an ELISA percentage (E%), calculated using the following formula: [E% = (sample Optical Density (OD)/mean OD of positive controls) × 100]. The positive threshold values were set as suggested by the manufacturers: sera with E% > 50 were considered positive.

### 2.3. Statistical Analysis

The individual apparent prevalence of antibodies against *C. burnetii* was estimated from the proportion of seropositive animals to the total number of individuals analyzed, using the two-sided exact binomial test, with 95% CI. To homogenize the scales of the explanatory variables, cut-off points for continuous variables were determined at the 33rd and 66th percentiles. Coefficients and standard error values generated using an intercept-only generalized estimating equation (GEE) binomial logistic regression model, with the hunting ground as the subject variable, were used to adjust the estimated seroprevalence and 95%CI for clustering at hunting ground level [23]. Pearson´s Chi-square or Fisher’s exact test was first used, as appropriate, to screen for associations between seroprevalence with explanatory variables. All variables with a *p* < 0.05 in the bivariate analysis were selected for further analyses. Collinearity between pairs of variables was then tested using Cramer’s V coefficient. When collinearity was detected (Cramer’s V coefficient ≥ 0.6), the variable with the strongest a priori biological association with *C. burnetii* was retained. Finally, a GEE analysis was carried out to study the effect of the variables selected from the bivariate analysis. The number of seropositive animals was assumed to follow a binomial distribution, and “hunting ground” was included as the subject variable. Forward selection was used for introduction of variables, starting with the variable with the lowest *p*-value in the bivariate analysis. At each step, the confounding effect of the included variable was assessed by calculating the change in odds ratio (OR). The model was re-run until all remaining variables showed statistically significant values (*p* < 0.05). For the choice of the best model, the quasi-likelihood under the independence model criterion (QIC) was considered. SPSS 25.0 software (Statistical Package for Social Sciences, Inc., Chicago, IL, USA) was used for all statistical analyses.

### 2.4. Spatial Cluster Analysis

A spatial scan statistical analysis was applied using a Bernoulli model [24] to detect areas with significant aggregations of high seroprevalence at hunting ground level, using SaTScanTM v10.1.2 software. The number of Monte Carlo simulations was set to 1000 for the cluster scan statistic. SaTScan was used to estimate relative risk (RR), representing the relative frequency of seropositive individuals compared to baseline, for each cluster. Clusters were considered significant at *p* < 0.05.

## 3. Results and Discussion

In the present study, we detected moderate and endemic circulation of *C. burnetii* in wild lagomorph populations of Spanish Mediterranean ecosystems. The overall apparent individual seroprevalence was 8.9% (57/638; 95% CI: 6.8–11.4) (Table 1). By species, antibodies against *C. burnetii* were found in 11.3% (53/471; 95% CI: 8.4–14.1) of European wild rabbits and 2.4% (4/167; 95% CI: 0.1–4.7) of Iberian hares. After adjustment for clustering, the estimated individual seroprevalences were 12.9% (95% CI: 7.2–22.3) in wild rabbit and 2.5% (95% CI: 0.9–6.5) in Iberian hare, which denote different exposure level to *C. burnetii* between these two lagomorph species. This finding could be related to differences in behavior. While Iberian hares are generally solitary [25], wild rabbits live in social groups in burrows [26], which might favor the transmission and maintenance of the bacterium. 

The prevalence of anti-*C. burnetii* antibodies detected in wild rabbits in the present study is lower than those reported in the only two previous serosurveys conducted on this species so far. A seroprevalence of 37.9% (176/464) was found in wild rabbit populations in a survey carried out in different regions of Spain, including Andalusia, where certain areas exhibited seroprevalence values ranging between 45.0% and 62.5% [27]. Higher seroprevalence (65.5%; 394/602) was also found in wild rabbits from central areas of this country [28]. The differences may be due to the presence of sympatric wild ungulates, particularly of red deer (Cervus elaphus) [28,29], which is considered an important node in the epidemiological cycle of this pathogen in Mediterranean ecosystems [27]. While both studies found higher seroprevalences in areas with a high density of this wild ruminant species, in our study, red deer were only present in 10 of the 112 (8.9%) hunting grounds analyzed. Nevertheless, a further large-scale serosurvey is warranted to assess differences among regions.

To the best of the authors’ knowledge, this is the first report of *C. burnetii* exposure in Iberian hare, increasing the range of susceptible species to this zoonotic pathogen. The low seroprevalence detected in this species is consistent with the absence of antibodies against *C. burnetii* reported in European hare (*Lepus europaeus*) in Germany (0/78) [30], the Czech Republic (0/48) [31] and Greece (0/105) [32]. In contrast, higher seropositivity values were observed in other hare species, including the black-tailed jack rabbit (*Lepus californicus*) in the USA (39.4%; 99/251) [33] and in snowshoe hares (*Lepus americanus*) in Canada (45.5%; 10/22) [7]. The seroprevalence detected in Iberian hare suggests that this species may be considered spillover hosts rather than true reservoirs for *C. burnetii* in the study area, although additional studies, including a high number of tested animals, are needed to support this hypothesis. 

Temporally, anti-*C. burnetii* antibodies were detected in all the hunting season samples and an increasing trend in seroprevalence was found over the years. The lowest seroprevalence (3.3%) was detected during the 2018–2019 hunting season, rising to 20.9% during 2021/2022. Geographically, seropositive animals were detected in all provinces with frequencies of antibodies ranging between 0.9% (1/107; 95% CI: 0.0–2.8) in Córdoba and 46.5% (33/71; 95% CI: 34.9–58.1) in Cádiz, indicating a wide but heterogeneous spatial distribution of *C. burnetii* in wild lagomorphs in the Mediterranean ecosystems of southern Spain. Indeed, the GEE model revealed that the prevalence of anti-*C. burnetii* antibodies was significantly higher (*p* = 0.006) in western (20.7%) compared to central (1.5%) Andalusia. Consistently, spatial analysis identified one statistically significant cluster of hunting grounds positive for anti-*C. burnetii* antibodies in southwestern Andalusia (RR: 13.6; *p* < 0.001) (Figure 1). Previous studies have suggested that host density could be associated with a higher risk of *C. burnetii* exposure in mammal species, including wild lagomorphs [27,28,34]. In this respect, a higher mean of hunted wild lagomorphs, which is a reliable index of relative animal abundance [35], was reported in the western region of Andalusia compared with central and eastern regions [36]. In addition, the western region presents higher mean humidity and less-extreme mean temperatures than those observed in central and eastern regions [19]. In fact, the area identified in the spatial cluster includes the Spanish area with the highest average annual rainfall [19]. These climatic features may favor not only the persistence of the bacterium in pastures but also the presence and abundance of competent tick species in the environment. All these findings indicate that western Andalusia could be a hotspot area for *C. burnetii* circulation.

The presence of sheep (*Ovis aries*) in the sampled hunting grounds was also identified as a risk factor potentially associated with *C. burnetii* exposure (Table 2), which is consistent with previous studies carried out on domestic ruminants [37]. Hunting grounds where sheep were present had 4.6 times more risk of *C. burnetii* exposure than those without sheep. Domestic ruminants are recognized as the main reservoirs of *C. burnetii* and sheep are frequently raised under extensive production systems in the study region [38]. In this respect, pastures contaminated with feces from *C. burnetii* positive livestock have been described as an important source for infection in wildlife [33], including wild rabbit and hare species [8,33,39]. This hypothesis is supported by the high seroprevalence of *C. burnetii* detected by our research group in sheep farms of Andalusia, where 100.0% of the farms had at least one positive sheep and 40.0% of the tested sheep were confirmed to be exposed to *C. burnetii* (unpublished data). 

## 4. Conclusions

In summary, the results obtained in the present study revealed widespread, but not homogeneous exposure to *C. burnetii* in wild lagomorph populations of Spanish Mediterranean ecosystems. The risk factors identified (western Andalusia and hunting grounds with presence of sheep) should be prioritized in future risk-based surveillance programs for *C. burnetii*. Additional molecular and serological studies are required to elucidate differences between wild rabbits and Iberian hares and to assess the risk of zoonotic transmission of *C. burnetii* from these wild lagomorph species in Iberian Mediterranean ecosystems.

## Figures and Tables

**Figure 1 animals-14-00749-f001:**
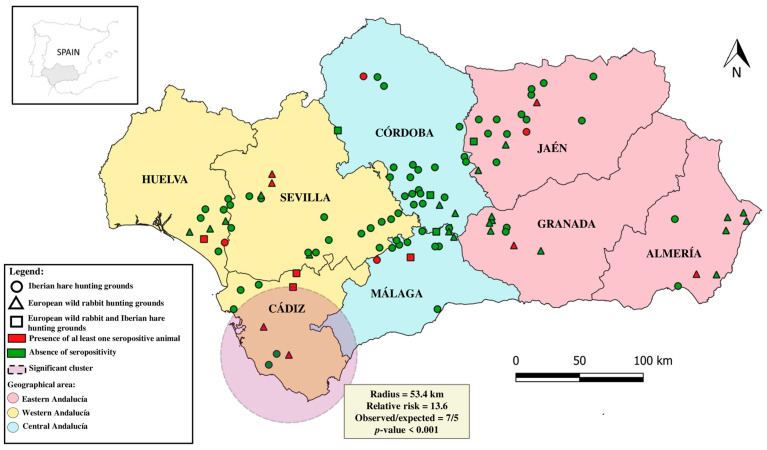
Spatial distribution of *C. burnetii* seroprevalence in wild lagomorphs in the study region (Andalusia, southern Spain). The discontinuous circle represents a significant cluster of seropositivity identified using spatial analysis.

**Table 1 animals-14-00749-t001:** Distribution of the seroprevalence against *Coxiella burnetii* in wild lagomorphs in Andalusia (southern Spain) by animal and hunting ground categories and results of the bivariate analysis.

Variable	Categories	No. Positives/ Overall ^1^	Seroprevalence (%)	*p*
**Data recorded from the sampled animals**				
Species	Wild rabbit	53/471	11.3	**<0.001**
	Iberian hare	4/167	2.4	
Age	Adult	53/460	11.5	**0.001**
	Subadult	3/135	2.2	
	Young	0/36	0.0	
Sex	Male	28/307	9.1	0.516
	Female	29/326	8.9	
Kidney fat index	0	19/154	12.3	0.541
	1	16/148	10.8	
	2	10/114	8.8	
	3	7/99	7.1	
Bodyweight (kg)	0.4–0.9	13/166	7.8	0.420
	1.0–1.2	7/144	4.9	
	1.3–3.1	7/147	4.8	
Body length (cm)	19–37	6/161	3.7	0.075
	38–40	14/143	9.8	
	41–59	7/136	5.1	
Hunting season	2018/2019	2/60	3.3	**<0.001**
	2019/2020	2/48	4.2	
	2020/2021	24/391	6.1	
	2021/2022	29/139	20.9	
**Hunting ground’s characteristics**				
Geographical area	Western	45/217	20.7	**<0.001**
	Central	3/196	1.5	
	Eastern	9/225	4.0	
Burrow density	High	43/418	10.3	**<0.001**
	Medium	10/37	27.0	
	Low	0/76	0.0	
High abundance of ticks in the hunting ground	Yes	24/297	8.1	**0.001**
	No	1/142	0.7	
High abundance of fleas in the hunting ground	Yes	16/273	5.9	0.514
	No	9/166	5.4	
Fenced hunting ground	Yes	5/37	13.5	0.305
	No	48/494	9.7	
Presence of rabbit feeders	Yes	26/310	8.4	0.097
	No	27/221	12.2	
Feed supplementation in rabbits	Yes	32/217	14.7	**0.002**
	No	21/314	6.7	
Presence of swamps	Yes	18/85	21.2	**0.001**
	No	35/446	7.8	
Presence of troughs	Yes	50/442	11.3	**0.012**
	No	3/89	3.4	
Presence of streams	Yes	23/277	8.3	0.115
	No	30/254	11.8	
The hunting ground is weeded	Yes	17/118	14.4	0.054
	No	36/413	8.7	
Presence of artificial burrows	Yes	4/37	10.8	0.517
	No	49/494	9.9	
**Detection of clinical cases of other infectious diseases**				
Outbreaks of myxomatosis in the last year	Yes	53/508	10.4	0.084
	No	0/23	0.0	
Outbreaks of RHD ^2^ in the last year	Yes	47/435	10.8	0.120
	No	6/96	6.3	
Outbreaks of myxomatosis in the last month	Yes	51/434	11.8	**0.001**
	No	2/97	2.1	
Outbreaks of RHD ^2^ in the last month	Yes	18/121	14.9	**0.034**
	No	35/410	8.5	
**Presence of other sympatric species in the hunting ground**				
Presence of wild boar (*Sus scrofa*)	Yes	18/333	5.4	**0.003**
	No	34/286	11.9	
Presence of red deer (*Cervus elaphus*)	Yes	3/63	4.8	0.199
	No	49/556	8.8	
Presence of wildcat (*Felis silvestris*)	Yes	14/170	8.2	0.224
	No	39/361	10.8	
Presence of Iberian lynx (*Lynx pardinus*)	Yes	14/86	16.3	**0.032**
	No	39/445	8.8	
Presence of domestic cat (*Felis silvestris catus*)	Yes	45/476	9.5	0.168
	No	8/55	14.5	
Presence of dog (*Canis familiaris*)	Yes	34/396	8.6	0.051
	No	19/135	14.1	
Presence of cattle (*Bos taurus*)	Yes	5/25	20.0	**0.012**
	No	19/372	5.1	
Presence of goat (*Capra aegagrus hircus*)	Yes	5/149	3.4	0.060
	No	19/248	7.7	
Presence of sheep (*Ovis aries*)	Yes	21/192	10.9	**<0.001**
	No	3/205	1.5	
Presence of farmed rabbit (*Oryctolagus cuniculus*)	Yes	7/60	11.7	0.390
	No	46/471	9.8	
Presence of domestic pig (*Sus scrofa domesticus*)	Yes	5/40	12.5	0.081
	No	19/357	5.3	
**Climate characteristics of the hunting ground**				
Mean temperature (°C)	12.3–16.8	2/230	0.9	**<0.001**
	16.9–17.4	7/106	6.6	
	17.5–18.5	17/163	10.4	
Max temperature (°C)	18.9–23.0	10/184	5.4	0.060
	23.1–24.2	8/218	3.7	
	24.3–27.4	8/73	11.0	
Mean annual rainfall (mm)	273.3–563.8	3/169	1.8	**0.002**
	563.9–597.9	9/199	4.5	
	598.0–1134.6	14/131	10.7	
Humidity (g/m^3^)	33–56	0/26	0.0	0.346
	57–65	9/132	6.8	
	66–100	6/75	8.0	

^1^ Missing values omitted, ^2^ RHD: rabbit hemorrhagic disease.

**Table 2 animals-14-00749-t002:** Generalized estimating equations analysis of risk factors associated with exposure to *Coxiella burnetii* in wild lagomorphs in Andalusia (southern Spain).

Variable	Categories	*p*-Value	OR 95% CI
Presence of sheep	Yes	0.023	4.6 (1.2–17.0)
	No	a	a
Geographical area	Western	0.006	19.9 (2.3–170.4)
	Eastern	0.133	5.2 (0.6–43.8)
	Central	a	a

a: Reference Category.

## Data Availability

The data presented in this study are available on request from the corresponding author. The data are not publicly available due to privacy restrictions and long extension of datasets.

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
