# Peer review of "Exposure to Coxiella burnetii in Wild Lagomorphs in Spanish Mediterranean Ecosystems"

_animals, 2024, doi:10.3390/ani14050749_

Round 1
Reviewer 1 Report
Comments and Suggestions for Authors
The ms "Exposure to Coxiella burnetii in wild lagomorphs in Spanish Mediterranean ecosystems" investigated the presence and circulation of the potentially zoonotic C. burnetii in European wild rabbits and Iberian hares in Andalucia, southern Spain, by means of serosurveys, to define the possible roles of these species as reservoirs/spillover hosts. The research is clearly and concisely presented, conclusions are supported by results, and the paper only needs minor revision before being publishable.
Find below some suggestions:
Line 92: the sentence could be clearer
Line 98: remove "kindly"
Line 104: "heart"
Line 105: "stored at..."
Line 164: replace with "we detected moderate and endemic circulation..."
Authors might mention as Further research proposed and recommended the attempt to molecularly detect pathogen presence in rabbits/hare samples by PCR, also to compare to serosurveys results, in order to be able to characterize C. burnetii strains circulating in those species/areas. They might also briefly discuss the particularly high prevalence of seropositive animals found in Cadiz.
The ms could also be presented as a Short Communication
Comments on the Quality of English Language
English language is fine throughout the ms, just needs minor revisions
Author Response
Answer to Reviewer’s 1 Comments:
The ms "Exposure to Coxiella burnetii in wild lagomorphs in Spanish Mediterranean ecosystems" investigated the presence and circulation of the potentially zoonotic C. burnetii in European wild rabbits and Iberian hares in Andalucia, southern Spain, by means of serosurveys, to define the possible roles of these species as reservoirs/spillover hosts. The research is clearly and concisely presented, conclusions are supported by results, and the paper only needs minor revision before being publishable.
We would like to thank the reviewer for their positive feedback and for the comments on how the manuscript can be improved.
Find below some suggestions:
Line 92: the sentence could be clearer
Following the reviewer’s suggestion, the sentence has been rephrased as “The sample size was determined based on an assumed prevalence of 50%, which provides the maximum sample size in studies where the prevalence is unknown. The calculation was done with a 95% confidence interval (95%CI) and a desired precision of ±5%”.
Line 98: remove "kindly"
Done.
Line 104: "heart"
Thanks for spotting this mistake, the term has been corrected.
Line 105: "stored at..."
Done.
Line 164: replace with "we detected moderate and endemic circulation..."
Done.
Authors might mention as Further research proposed and recommended the attempt to molecularly detect pathogen presence in rabbits/hare samples by PCR, also to compare to serosurveys results, in order to be able to characterize C. burnetii strains circulating in those species/areas.
Following the reviewer’s suggestion, we have added a sentence in the conclusion section (L. 251-254)
They might also briefly discuss the particularly high prevalence of seropositive animals found in Cadiz.
Please note that we stated in current lines 221-224 that “In fact, the area identified in the spatial cluster includes the Spanish area with the highest average annual rainfall. These climatic features may favor not only the persistence of the bacterium in pastures but also the presence and abundance of competent tick species in the environment”. The area identified in the spatial cluster is located in Cadiz, explaining the high prevalence of seropositive animals in this province.
The ms could also be presented as a Short Communication
We sincerely think the current work contains results significant enough to warrant maintaining its current format. Converting it into a short communication may limit the ability to thoroughly discuss the results.
Reviewer 2 Report
Comments and Suggestions for Authors
The manuscript submitted by Castro-Scholten et al. provides compelling evidence of the exposure of European rabbits and Iberian hares to Coxiella burnetii, the causative agent of Q fever.
Overall, the study is meticulously planned and well-written, and I thoroughly enjoyed reading it. However, I would like to offer some suggestions to enhance the manuscript further:
-
Lines 156-163: In my view, this information should be relocated to the introduction to lend greater weight and assertiveness to the necessity of conducting this study.
-
Line 165: Before stating that this is the first report of exposure to C. burnetii in Iberian hares, consider describing the seroprevalence results (Lines 168-177) to better organize the beginning of the Results and Discussion section.
-
Table 1: Be careful with the superscript in "Outbreaks of RHD2 in the last month."
-
Lines 184-185: While reference 27 encompasses several regions of the Iberian Peninsula with an average seroprevalence of 37.9%, it would be valuable to specify that certain areas in Andalusia exhibited a seroprevalence exceeding 50%.
-
Line 192: Towards the end of this paragraph, consider adding the relationship that the authors have identified between the seroprevalence of C. burnetii and the presence of other non-ungulated species, particularly the Iberian lynx. Were these relationships described before? Are there prior studies on lynx prevalence or exposure to C. burnetii? What about wild boars?
-
Lines 203-205: Given that the authors note the highest seroprevalence in the 2021/2022 season and its correlation with climatological conditions, explore whether the increase could be attributed to optimized environmental conditions facilitating greater bacterial circulation. Are there alternative causes?
-
Lines 217-220: In Table 1, the authors highlight a connection between seroprevalence and the presence of swamps and canals. It would be insightful to confirm if the higher seroprevalence in western Andalusia is not solely due to its climate but also the presence of substantial bodies of water.
-
Line 243: I would encourage you to discuss the relationship found between seroprevalence and outbreaks of myxomatosis and RHD in the month preceding sampling. Are there existing studies linking these diseases with C. burnetii?
Finally, although the current manuscript's content is sufficiently relevant, I recommend the authors consider complementing their findings with molecular analysis in the future, specifically in the search for C. burnetii.
Author Response
Answer to Reviewer’s 2 Comments:
The manuscript submitted by Castro-Scholten et al. provides compelling evidence of the exposure of European rabbits and Iberian hares to Coxiella burnetii, the causative agent of Q fever. Overall, the study is meticulously planned and well-written, and I thoroughly enjoyed reading it.
We thank the reviewer for her/his preliminary appraisal of our manuscript.
However, I would like to offer some suggestions to enhance the manuscript further:
Lines 156-163: In my view, this information should be relocated to the introduction to lend greater weight and assertiveness to the necessity of conducting this study.
Following the reviewer’s suggestion, the sentence has been moved to the introduction section (L. 61-63 and L. 83-88).
Line 165: Before stating that this is the first report of exposure to C. burnetii in Iberian hares, consider describing the seroprevalence results (Lines 168-177) to better organize the beginning of the Results and Discussion section.
Following the reviewer’s suggestion, we have relocated this statement after describing the seroprevalence results (L. 193-194).
Table 1: Be careful with the superscript in "Outbreaks of RHD2 in the last month."
Thanks for spotting this mistake, the superscript has been corrected.
Lines 184-185: While reference 27 encompasses several regions of the Iberian Peninsula with an average seroprevalence of 37.9%, it would be valuable to specify that certain areas in Andalusia exhibited a seroprevalence exceeding 50%.
Following the reviewer’s suggestion, we included additional information about seroprevalences detected in Andalusia (L. 183-184).
Line 192: Towards the end of this paragraph, consider adding the relationship that the authors have identified between the seroprevalence of C. burnetii and the presence of other non-ungulated species, particularly the Iberian lynx. Were these relationships described before? Are there prior studies on lynx prevalence or exposure to C. burnetii? What about wild boars?
In the present study, the explanatory variables “presence of Iberian lynx” and “wild boar” as well as 15 other independent variables with p-value < 0.05 in bivariate analysis were selected for inclusion in the multivariate analysis after assessing collinearity. Although we do agree that it would be interesting to discuss all the significant variables that were not retained in the final GEE model, we consider that, because of the large number of variables that met this criterion, including discussion about each of them is likely to divert attention away from the focus of the results and confuse the reader. Moreover, the relationship between exposure to C. burnetii in wild lagomorphs and the presence of Iberian lynx has not been previously described and there are no studies on this bacterium in this feline species. Therefore, we agree with the reviewer that it would be of interest to carry out epidemiological studies of C. burnetii in the Iberian lynx. In addition, no studies have assessed the relationship between seropositivity to C. burnetii in wild lagomorphs and the presence of wild boar. Indeed, there is very limited information available on the presence of C. burnetii in wild boar from Iberian Peninsula (Espí et al., 2021; Zendoia et al., 2022; Pires et al., 2023).
Lines 203-205: Given that the authors note the highest seroprevalence in the 2021/2022 season and its correlation with climatological conditions, explore whether the increase could be attributed to optimized environmental conditions facilitating greater bacterial circulation. Are there alternative causes?
After reviewing the climatological data recorded during the study period (https://www.aemet.es/es/serviciosclimaticos/datosclimatologicos), we did not detect variations in the climatic conditions that could increase the risk to be exposed to C. burnetii over these years. In addition, it should be noted that sampling was not homogeneous across the study geographical area during hunting seasons. Thus, the variable “hunting season” was correlated with “geographical area”. The number of animals sampled from western Andalusia was highest during the hunting season 2021/2022.
Lines 217-220: In Table 1, the authors highlight a connection between seroprevalence and the presence of swamps and canals. It would be insightful to confirm if the higher seroprevalence in western Andalusia is not solely due to its climate but also the presence of substantial bodies of water.
We agree with the reviewer that the higher seroprevalence in western Andalusia could be due not only to more humid weather conditions but also to other factors. Nevertheless, according to the date collected through the epidemiological questionnaire during sampling, the number of hunting grounds with presence of swamps was not significantly higher in western compared to central and eastern Andalusia.
Line 243: I would encourage you to discuss the relationship found between seroprevalence and outbreaks of myxomatosis and RHD in the month preceding sampling. Are there existing studies linking these diseases with C. burnetii?
As mentioned above, outbreak of myxomatosis and RHD in the month preceding sampling were not retained in the GEE analysis. For this reason, we do not discuss these differences in the discussion. In addition, although a relationship between myxomatosis and RHD has been established with other pathogens, such as Toxoplasma gondii (Mason et al., 2015) and Eimeria stiedae (Boag et al., 2013), to date there are no studies linking myxomatosis or RHD with C. burnetii.
Finally, although the current manuscript's content is sufficiently relevant, I recommend the authors consider complementing their findings with molecular analysis in the future, specifically in the search for C. burnetii.
We agree with the reviewer that molecular analysis would enhance our findings, so further studies in this research line are warranted. Following the reviewer’s suggestion, a new sentence has been included in the conclusion section (L. 251-254).